# *PON1* Q192R is associated with high platelet reactivity with clopidogrel in patients undergoing elective neurointervention: A prospective single-center cohort study

Koji Tanaka[1], Shoji Matsumoto[2¤a], Gulibahaer Ainiding[1], Ichiro Nakahara[2¤a], Hidehisa Nishi[2], Tetsuya Hashimoto[2], Tsuyoshi Ohta[2¤b], Nobutake Sadamasa[2], Ryota Ishibashi[2], Masanori Gomi[2], Makoto Saka[2], Haruka Miyata[2], Sadayoshi Watanabe[2], Takuya Okata[2], Kazutaka Sonoda[2], Junpei Koge[2], Kyoko M. Iinuma[1], Konosuke Furuta[3], Izumi Nagata[2], Keitaro Matsuo[4,5], Takuya Matsushita[1], Noriko Isobe[1], Ryo Yamasaki[1], Jun-ichi Kira[1¤c¤d]*

1 Department of Neurology, Graduate School of Medical Sciences, Neurological Institute, Kyushu University, Fukuoka, Japan, 2 Department of Neurosurgery, Kokura Memorial Hospital, Kitakyushu, Japan, 3 Department of Neurology, Kokura Memorial Hospital, Kitakyushu, Japan, 4 Department of Cancer Epidemiology, Nagoya University Graduate School of Medicine, Nagoya, Japan, 5 Division of Cancer Epidemiology and Prevention, Aichi Cancer Center Research Institute, Nagoya, Japan

¤a Current address: Department of Comprehensive Strokology, Fujita Health University School of Medicine, Toyoake, Japan
¤b Current address: Department of Neurosurgery, National Cerebral and Cardiovascular Center, Suita, Japan
¤c Current address: Translational Neuroscience Center, Graduate School of Medicine, and School of Pharmacy at Fukuoka, International University of Health and Welfare, Okawa, Japan
¤d Current address: Department of Neurology, Brain and Nerve Center, Fukuoka Central Hospital, International University of Health and Welfare, Fukuoka, Japan
* kira@neuro.med.kyushu-u.ac.jp

**Data Availability Statement:** All relevant data are within the manuscript and its Supporting Information files.

## Abstract

### Background and purpose

The impact of the paraoxonase-1 (*PON1*) polymorphism, Q192R, on platelet inhibition in response to clopidogrel remains controversial. We aimed to investigate the association between carrier status of *PON1* Q192R and high platelet reactivity (HPR) with clopidogrel in patients undergoing elective neurointervention.

### Methods

Post-clopidogrel platelet reactivity was measured using a VerifyNow® P2Y12 assay in P2Y12 reaction units (PRU) for consecutive patients before the treatment. Genotype testing was performed for *PON1* Q192R and *CYP2C19*2* and *\*3* (no function alleles), and *\*17*. PRU was corrected on the basis of hematocrit. We investigated associations between factors including carrying ≥1 *PON1* 192R allele and HPR defined as original and corrected PRU ≥208.

**Funding:** SM has received research support from Japan Society for the Promotion of Science Grant-in-Aid for Scientific Research, grant number JP22591591. The funders had no role in study design, data collection and analysis, decision to publish, or preparation of the manuscript.

## Results

Of 475 patients (232 men, median age, 68 years), HPR by original and corrected PRU was observed in 259 and 199 patients (54.5% and 41.9%), respectively. Carriers of ≥1 *PON1* 192R allele more frequently had HPR by original and corrected PRU compared with non-carriers (91.5% vs 85.2%, P = 0.031 and 92.5% vs 85.9%, P = 0.026, respectively). In multi-variate analyses, carrying ≥1 *PON1* 192R allele was associated with HPR by original (odds ratio [OR] 1.96, 95% confidence interval [CI] 1.03–3.76) and corrected PRU (OR 2.34, 95% CI 1.21–4.74) after adjustment for age, sex, treatment with antihypertensive medications, hematocrit, platelet count, total cholesterol, and carrying ≥1 *CYP2C19* no function allele.

## Conclusions

Carrying ≥1 *PON1* 192R allele is associated with HPR by original and corrected PRU with clopidogrel in patients undergoing elective neurointervention, although alternative results related to other genetic polymorphisms cannot be excluded.

## Introduction

Dual antiplatelet therapy (DAPT), most commonly with aspirin and clopidogrel, is regularly used for antiplatelet premedication to prevent thromboembolic complications following endo-vascular treatment. However, there is marked inter-patient variability of platelet inhibition in response to clopidogrel, and high platelet reactivity (HPR) with clopidogrel is associated with an increased risk of thromboembolic complications following endovascular treatment [1–3].

Clopidogrel is a thienopyridine that blocks the P2Y12 receptor. The antiplatelet activity of clopidogrel depends on its conversion to an active metabolite mainly by the cytochrome P450 family 2 subfamily C member 19 (CYP2C19). *CYP2C19* is a highly polymorphic gene. *CYP2C19*\**2* and \**3* were reported to be the most common single-nucleotide polymorphisms (SNPs) that decreased enzyme activity in a multi-ethnic population. Carrying ≥1 *CYP2C19* no function allele was associated with HPR with clopidogrel and thromboembolic complications following endovascular treatment [4–6]. Conversely, *CYP2C19*\**17* is an increased function SNP that contributes to enhanced platelet inhibition in response to clopidogrel [7].

Paraoxonase-1 (PON1) is another enzyme that participates in the second step of clopidogrel metabolism, and the *PON1* Q192R SNP is a major determinant of PON1 activity [8]. Com-pared with carriers of ≥1 *PON1* 192R allele, non-carriers have lower PON1 activity and lower platelet inhibition in response to clopidogrel, which leads to a higher risk of stent thrombosis following coronary artery stenting [8]. However, unlike *CYP2C19* SNPs, the impact of *PON1* Q192R on platelet inhibition in response to clopidogrel remains controversial; the association between *PON1* Q192R and HPR with clopidogrel has not been replicated in other studies [9–12]. Conversely, recent studies have shown that carriers of ≥1 *PON1* 192R allele have reduced platelet inhibition in response to clopidogrel compared with non-carriers [13–16].

A recent study demonstrated that HPR also led to thromboembolic complications in patients undergoing pipeline embolization device placement for intracranial aneurism (IA) [17]. However, to date, little is known about the association between *PON1* Q192R and HPR with clopidogrel in patients undergoing neurointervention [18]. Therefore, this study investi-gated the impact of *PON1* Q192R on HPR with clopidogrel in patients undergoing elective neurointervention.

## Materials and methods

### Ethics statement

The ethics committee at Kyushu University Hospital (389–01) and Kokura Memorial Hospital approved data collection from patients undergoing elective neurointervention at Kokura Memorial Hospital and genotype testing at Kyushu University. Written informed consent was obtained from all participants.

### Study patients

Five hundred and sixty-seven patients who underwent scheduled endovascular treatment for cerebrovascular disease from May 2010 to September 2015 were enrolled in this observational study. All patients were on DAPT before neurointervention. Alongside low-dose acetylsalicylic acid therapy, 75 mg clopidogrel was administered for $\geq$7 days before treatment. For patients administered clopidogrel <7 days before treatment, clopidogrel was administered at a loading dose of 300 mg followed by 75 mg daily. The following clinical information was systematically extracted from medical records: age, sex, body mass index, vascular risk factors (hypertension, diabetes mellitus, dyslipidemia, and current smoking status), ischemic heart disease, and concomitant treatment with antihypertensive medications (calcium channel blockers, beta blockers, and/or angiotensin-converting enzyme inhibitors [ACEI]/angiotensin receptor blockers [ARB]) and proton pump inhibitors. Laboratory data, including hematocrit, platelet count, and serum cholesterol levels were obtained from routine laboratory testing principally performed a few days before treatment.

### Platelet function measurements

Blood samples (1.8 ml) were taken from the median cubital vein in vacuum collection tubes containing 0.2 ml of 3.2% sodium citrate using a 21-G blood collection needle before the procedure on the day of neurointervention. Platelet reactivity was evaluated using a VerifyNow® P2Y12 assay (Accumetrics, San Diego, CA, USA) according to the manufacturer's instructions. Platelet reactivity was expressed as P2Y12 reaction units (PRU). Because hematocrit alters VerifyNow P2Y12 assay results, we calculated the corrected PRU by subtracting 7.5 PRU for every % of hematocrit below and adding 7.5 PRU for every % of hematocrit above, 42% as previously proposed [19]. We defined HPR as an original PRU of $\geq$208 [20], as well as a corrected PRU of $\geq$208.

### DNA extraction and genotyping

Genomic DNA was extracted from 200 μl of whole blood collected in ethylenediaminetetraacetate-coated tubes using a NucleoSpin® Blood QuickPure kit (Macherey-Nagel, Düren, Germany) according to the manufacturer's instructions. To evaluate the quantity and quality of extracted DNA, the optical density was measured using a NanoDrop™ 1000 Spectrophotometer (Thermo Fisher Scientific, Wilmington, DE, USA). The DNA concentration was adjusted to 10 ng/μl before assays were performed.

Genotyping was performed using the TaqMan™ Drug Metabolism Genotyping Assay (Applied Biosystems, Foster City, CA, USA) for the following SNPs: *PON1* Q192R (rs662; c.575A>G rs662 or p.Gln192Arg), *CYP2C19*2* (rs4244285; c.681G>A or p.Pro227Pro), *CYP2C19*3* (rs4986893; c.636G>A or p.Trp212Ter), and *CYP2C19*17* (rs12248560; c.-806C>T). The total reaction volume (25 μl) consisted of 2 × TaqMan Universal Polymerase Chain Reaction Master Mix II (Applied Biosystems), 20 × primer and probe mix (TaqMan Drug Metabolism Genotyping Assay Mix), and 20 ng of template DNA according to the

manufacturer's instructions. Polymerase chain reaction was performed using the 7500 Real-Time PCR System (Applied Biosystems). Allele discrimination was manually and automatically performed using the 7500 System Sequence Detection software (Applied Biosystems).

## Statistical analysis

All statistical analyses were performed with JMP statistical software, version 9.0 (SAS Institute, Cary, NC, USA). Data are expressed as medians and interquartile ranges for continuous variables and counts and percentages for categorical variables. All SNPs were tested for deviation from Hardy–Weinberg equilibrium. Clinical characteristics and carrier status of at least one variant allele of Q192R of *PON1* and no function (*2 and *3) and increased function (*17) alleles of *CYP2C19* were compared between patients with and without HPR by original and corrected PRU using the Chi-squared test, Fisher's exact test, or the Wilcoxon rank-sum test as appropriate. A multivariate logistic regression analysis was performed to investigate factors associated with HPR by original and corrected PRU using forced entry and stepwise selection procedures. Age, sex, and carrier status of ≥1 *CYP2C19* no function allele and ≥1 *PON1* 192R allele were forced in; other variables were chosen by stepwise selection with a significance level of $\alpha = 0.10$ for entry and $\alpha = 0.10$ for removal. A P-value $< 0.05$ was considered statistically significant.

## Results

Of 567 patients who underwent scheduled endovascular treatment and were registered during the study period, 92 were excluded because of no clopidogrel use (n = 6), initiation of clopidogrel <7 days without a loading dose (n = 14), lack of PRU data (n = 37), or lack of genomic data (n = 35). Finally, 475 patients (232 male; median age, 68 years) were included in the following analysis. Two hundred and seventy-six patients received simple/balloon or stent-assisted coil embolization for ruptured (n = 13) or unruptured (n = 263) IA, 197 received stent placement for symptomatic (n = 117) or asymptomatic (n = 80) carotid artery stenosis, 1 received embolization for carotid-cavernous fistula, and 1 received parent artery occlusion.

Results from genetic testing are shown in Table 1. Four hundred and twenty-one patients (88.6%) carried ≥1 *PON1* 192R allele. Two hundred and fifty-two patients (53.1%) carried ≥1 *CYP2C19*2 allele and 105 patients (22.1%) carried ≥1 *CYP2C19*3 allele. Thus, 321 patients

**Table 1. Genotype distribution among the study population.**

| Alleles | | Total (n = 475) | MAF (%) | Hardy-Weinberg equilibrium (*P* value) | MAF from the 1000 Genomes Project (East Asian, %) |
|---|---|---|---|---|---|
| *PON1* Q192R | A/A | 54 (11.4) | 64.9 | 0.421 | 66.6 |
| (rs662, c.575A>G) | A/G | 225 (47.4) | | | |
| | G/G | 196 (41.3) | | | |
| *CYP2C19*2 | G/G | 223 (46.9) | 30.7 | 0.333 | 31.2 |
| (rs4244285, c.681G>A) | G/A | 212 (44.6) | | | |
| | A/A | 40 (8.4) | | | |
| *CYP2C19*3 | G/G | 370 (77.9) | 12.1 | 0.194 | 5.6 |
| (rs4986893, c.636G>A) | G/A | 95 (20.0) | | | |
| | A/A | 10 (2.1) | | | |
| *CYP2C19*17 | C/C | 472 (99.4) | 0.3 | 1.000 | 1.5 |
| (rs12248560, c.-806C>T) | C/T | 3 (0.6) | | | |
| | T/T | 0 (0) | | | |

Data are presented as n (%). MAF, minor allele frequency.

(67.6%) carried ≥1 *CYP2C19* no function allele. Three patients (0.6%) had the *CYP2C19*\**17* allele. All tested SNPs were in Hardy–Weinberg equilibrium. In this study, the minor allele frequency of *CYP2C19*\**3* was higher compared with the East Asian population assessed in the 1000 Genomes Project (12.1% vs 5.6%) [21].

HPR by original and corrected PRU of ≥208 was observed in 259 and 199 patients (54.5% and 41.9%), respectively. After the correction of PRU on the basis of hematocrit, 70 patients with HPR were re-classified as non-HPR and 10 patients without HPR were re-classified as HPR. The re-classification rate was 16.8% and the correction of PRU reduced the prevalence of HPR by 23.2%. There were 17 patients without the *PON1* 192R allele or *CYP2C19* no function allele, 137 carrying ≥1 *PON1* 192R allele but not the *CYP2C19* no function allele, 37 carrying ≥1 *CYP2C19* no function allele but not the *PON1* 192R allele, and 284 carrying ≥1 *PON1* 192R allele and the *CYP2C19* no function allele. Among them, the frequencies of HPR by original PRU were 17.7%, 38.7%, 51.4%, and 64.8%, respectively, and those by corrected PRU were 11.8%, 23.4%, 35.1%, and 53.5%, respectively.

Patient characteristics are presented in Table 2. Patients with HPR by original PRU were more frequently female (59.1% vs 41.7%, respectively; P < 0.001) with lower platelet counts (median, $19.9 \times 10^4$/μl vs $21.0 \times 10^4$/μl, respectively; P < 0.001), and hematocrit levels (median, 38.7% vs 40.5%, respectively; P < 0.001) compared with those without HPR. Compared with non-carriers, carriers of ≥1 *PON1* 192R allele (91.5% vs 85.2%, respectively; P = 0.031) and ≥1 *CYP2C19* no function allele (78.4% vs 54.6%, respectively; P < 0.001) more frequently had HPR by original PRU. A similar trend was observed when comparing patients with and

**Table 2. Univariate analysis of high platelet reactivity in patients undergoing elective neurointervention.**

| Variables | Total (n = 475) | HPR by original PRU | | | HPR by corrected PRU | | |
|---|---|---|---|---|---|---|---|
| | | HPR (n = 259) | Non-HPR (n = 216) | *P* value | HPR (n = 199) | Non-HPR (n = 276) | *P* value |
| Sex, male | 232 (48.8) | 106 (40.9) | 126 (58.3) | <0.001 | 86 (43.2) | 146 (52.9) | 0.037 |
| Age (years) | 68 (58–75) | 68 (59–75) | 67 (57–75) | 0.336 | 68 (59–74) | 68 (58–76) | 0.862 |
| Body mass index | 22.9 (21.0–25.0) | 22.8 (20.8–24.9) | 23.0 (21.1–25.2) | 0.483 | 23.2 (21.0–25.2) | 22.5 (21.0–24.6) | 0.128 |
| Hypertension | 284 (59.8) | 159 (61.4) | 125 (57.9) | 0.436 | 122 (61.3) | 162 (58.7) | 0.567 |
| Dyslipidemia | 269 (56.6) | 144 (55.6) | 125 (57.9) | 0.619 | 113 (56.8) | 156 (56.5) | 0.955 |
| Diabetes mellitus | 113 (23.8) | 58 (22.4) | 55 (25.5) | 0.434 | 45 (22.6) | 68 (24.6) | 0.609 |
| Ischemic heart disease | 91 (19.2) | 49 (18.9) | 42 (19.4) | 0.885 | 33 (16.6) | 58 (21.0) | 0.226 |
| Current smoking | 56 (11.8) | 24 (9.3) | 32 (14.8) | 0.062 | 20 (10.1) | 36 (13.0) | 0.318 |
| Calcium channel blocker | 230 (48.4) | 134 (51.7) | 96 (44.4) | 0.113 | 101 (50.8) | 129 (46.7) | 0.388 |
| β-blocker | 57 (12.0) | 31 (12.0) | 26 (12.0) | 0.982 | 21 (10.6) | 36 (13.0) | 0.410 |
| ACEI/ARB | 177 (37.3) | 102 (39.4) | 75 (34.7) | 0.296 | 78 (39.2) | 99 (35.9) | 0.459 |
| Proton pump inhibitor | 351 (73.9) | 194 (74.9) | 157 (72.7) | 0.584 | 148 (74.4) | 203 (73.6) | 0.841 |
| Clopidogrel <7 days | 74 (15.6) | 45 (17.4) | 29 (13.4) | 0.237 | 34 (17.1) | 40 (14.5) | 0.442 |
| Platelet count ($\times 10^4$/μl) | 20.5 (17.3–23.8) | 19.9 (16.5–23.1) | 21.0 (18.4–24.9) | <0.001 | 19.9 (16.9–22.7) | 21.0 (18.0–25.0) | <0.001 |
| Hematocrit (%) | 39.4 (36.5–42.2) | 38.7 (35.7–40.9) | 40.5 (38.0–43.5) | <0.001 | 39.7 (37.0–42.6) | 39.2 (36.2–42.0) | 0.118 |
| Total cholesterol (mg/dl) | 185 (158–211) | 188 (160–212) | 182 (157–210) | 0.389 | 190 (161–215) | 180 (155–209) | 0.024 |
| HDL-C (mg/dl) | 54 (47–65) | 55 (47–68) | 53 (46–63) | 0.064 | 56 (48–67) | 54 (46–63) | 0.074 |
| LDL-C (mg/dl) | 110 (87–127) | 110 (86–124) | 110 (87–128) | 0.688 | 110 (88–131) | 110 (86–125) | 0.298 |
| ≥1 *PON1* 192R allele | 421 (88.6) | 237 (91.5) | 184 (85.2) | 0.031 | 184 (92.5) | 237 (85.9) | 0.026 |
| ≥1 *CYP2C19* no function allele | 321 (67.6) | 203 (78.4) | 118 (54.6) | <0.001 | 165 (82.9) | 156 (56.5) | <0.001 |
| ≥1 *CYP2C19* increased function allele | 3 (0.6) | 3 (1.2) | 0 (0) | 0.255 | 3 (1.5) | 0 (0) | 0.073 |

Data are presented as n (%) or median (interquartile range). ACEI, angiotensin-converting enzyme inhibitor; ARB, angiotensin receptor blocker; HDL-C, high-density lipoprotein-cholesterol; HPR, high platelet reactivity; LDL-C, low-density lipoprotein-cholesterol; PRU, P2Y12 reaction unit.

without HPR by corrected PRU, except when comparing higher total cholesterol levels (median, 190 mg/dl vs 179 mg/dl, P = 0.024) between patients with HPR by corrected PRU or those without HPR.

In a multivariate analysis, carrying ≥1 *PON1* 192R allele (odds ratio [OR] 1.96, 95% confidence interval [CI] 1.03–3.76), male sex (OR 0.50, 95% CI 0.32–0.78), treatment with a calcium channel blocker (OR 1.69, 95% CI 1.11–2.59), hematocrit (OR 0.87, 95% CI 0.83–0.92, per 1% increase), platelet count (OR 0.92, 95% CI 0.88–0.96, per $1 \times 10^4$/μl increase), and carrying ≥1 *CYP2C19* no function allele (OR 4.17, 95% CI 2.67–6.60) were associated with HPR by original PRU. Similarly, carrying ≥1 *PON1* 192R allele was associated with HPR by corrected PRU (OR 2.34, 95% CI 1.21–4.74) after adjustment for age, sex, treatment with ACEI/ARB, hematocrit, platelet count, total cholesterol, and carrying ≥1 *CYP2C19* no function allele (Table 3).

## Discussion

This study clarified the associations between the carrier status of *PON1* Q192R and HPR with clopidogrel in patients undergoing elective neurointervention. Carrying ≥1 *PON1* 192R allele as well as ≥1 *CYP2C19* no function allele was associated with HPR, which occurred in approximately half of the patients with clopidogrel in the present study. The prevalence of HPR was markedly higher in patients carrying both ≥1 *PON1* 192R allele and the *CYP2C19* no function allele compared with noncarriers.

Previous studies reported the prevalence of HPR with clopidogrel on the basis of a PRU of ≥208 ranged from 18.3% to 57.1% [22,23]. Together with multiple clinical, hematological, and biochemical parameters, the *CYP2C19* no function allele was associated with the inter-patient variability of platelet inhibition in response to clopidogrel [4,24,25]. In the present study, more than two-thirds of patients were carriers of ≥1 *CYP2C19* no function allele, which is in

**Table 3. Multivariate analysis of high platelet reactivity in patients undergoing elective neurointervention.**

| Variables | OR (95% CI) | *P* value |
|---|---|---|
| HPR by original PRU | | |
| Sex, male | 0.50 (0.32–0.78) | 0.002 |
| Age (per 1-year increase) | 1.00 (0.98–1.02) | 0.854 |
| Treatment with calcium channel blocker | 1.69 (1.11–2.59) | 0.015 |
| Hematocrit (per 1% increase) | 0.87 (0.83–0.92) | <0.001 |
| Platelet count (per $1 \times 10^4$/μl increase) | 0.92 (0.88–0.96) | <0.001 |
| Total cholesterol (per 10 mg/dl increase) | 1.05 (0.99–1.11) | 0.082 |
| ≥1 *PON1* 192R allele | 1.96 (1.03–3.76) | 0.039 |
| ≥1 *CYP2C19* no function allele | 4.17 (2.67–6.60) | <0.001 |
| HPR by corrected PRU | | |
| Sex, male | 0.54 (0.34–0.83) | 0.005 |
| Age (per 1-year increase) | 1.00 (0.98–1.02) | 0.993 |
| Treatment with ACEI/ARB | 1.47 (0.97–2.26) | 0.073 |
| Hematocrit (per 1% increase) | 1.05 (1.00–1.11) | 0.032 |
| Platelet count (per $1 \times 10^4$/μl increase) | 0.92 (0.88–0.95) | <0.001 |
| Total cholesterol (per 10 mg/dl increase) | 1.06 (1.00–1.12) | 0.033 |
| ≥1 *PON1* 192R allele | 2.34 (1.21–4.74) | 0.011 |
| ≥1 *CYP2C19* no function allele | 4.53 (2.87–7.30) | <0.001 |

ACEI, angiotensin-converting enzyme inhibitor; ARB, angiotensin receptor blocker; CI, confidence interval; HPR, high platelet reactivity; OR, odds ratio; PRU, P2Y12 reaction unit.

accordance with the observation that the frequency of carriers of ≥1 *CYP2C19* no function allele is markedly higher in East Asian populations compared with Western populations [6,26].

Similar to a previous study [19], the current study showed a significant negative association between hematocrit and PRU. Correcting for hematocrit resulted in an 8% re-classification and 15.4% reduction in the prevalence of HPR using a PRU threshold of ≥208, which may more accurately identify patients that would benefit from alternative antiplatelet therapy. Compared with the previous study, the current study showed a larger re-classification rate of 16.8% and a reduction in the prevalence of HPR of 23.2%. These differences between studies might be related to the relatively lower hematocrit levels in the current study that might explain the trend for re-classification as non-HPR by corrected PRU. In addition to the significant impact of carrying ≥1 *CYP2C19* no function allele, this study revealed that carrying ≥1 *PON1* 192R allele was a contributing factor for HPR based on the original and corrected PRU with clopidogrel in patients undergoing elective neurointervention. About 85% of clopidogrel is metabolized by CES1 and the remaining 15% is metabolized into a biologically active thiol metabolite via two steps. First, monooxygenation of the thiophene ring produces 2-oxo-clopidogrel. Second, oxidative cleavage of the thiolactone ring of 2-oxo-clopidogrel yields sulfenic acid, which is subsequently reduced to a thiol, an active metabolite bearing an exocyclic double bond (Fig 1). CYP2C19 contributes to both steps in the clopidogrel bioactivation pathway—the formation of 45% of 2-oxo-clopidogrel and 21% of the active metabolite [27]—whereas the PON1-mediated hydrolysis of 2-oxo-clopidogrel generates an endo metabolite, in which the double bond migrates from an exocyclic to an endocyclic position in the piperidine ring [28]. Unlike the active metabolite, the endo metabolite is unstable and ineffective for platelet inhibition [29]. These findings suggest that carriers of ≥1 *PON1* 192R allele have higher PON1 activity that accelerates the increase in the endo metabolite of clopidogrel compared with non-carriers. Moreover, lower CYP2C19 activity might further reinforce this cascade via the delayed metabolism of 2-oxo-clopidogrel by CYP2C19, which allows PON1 to metabolize 2-oxo-clopidogrel into the endo metabolite. Therefore, compared with Western populations, the impact of *PON1* Q192R on HPR with clopidogrel might be more apparent in East Asian populations, in which carriers of *CYP2C19* no function alleles are common. This study showed a markedly higher prevalence of HPR by original and corrected PRU in patients carrying ≥1 *PON1* 192R allele and the *CYP2C19* no function allele compared with noncarriers. Although no significant association between *PON1* polymorphisms and response to clopidogrel was reported by Saiz-Rodríguez et al. [18], this might be due to ethnic differences or combined assessment of the *PON1* polymorphisms, *PON1* Q192R, L55M (rs854560; c.163T>A or p. Leu55Met), and C-108T (rs705379; c.-108C>T). Our results are consistent with previous studies [13–16] but were opposite to those of Bouman et al. [8]. This suggests a difference in the role of PON1 between Western and East Asian populations. PON1 contributes to the antioxidative function of high-density lipoprotein-cholesterol, which potentially suppresses platelet aggregation [30]. Therefore, in addition to *CYP2C19* and *PON1* polymorphisms, differences in lipoprotein levels and their association with vascular disease risk might explain these results [31]. Our findings may have clinical utility in that *PON1* and *CYP2C19* genotyping may provide prognostic information on the efficacy of clopidogrel in patients undergoing neurointervention. Further study is needed to investigate the relative risk of thromboembolic complications between the carrier status of *PON1* 192R allele and *CYP2C19* no function allele, the benefit of alternative antiplatelet therapy with ticagrelor or prasugrel in patients carrying ≥1 *PON1* 192R allele and the *CYP2C19* no function allele, and whether the de-escalation of DAPT to clopidogrel monotherapy can avoid more bleeding events in non-carriers.

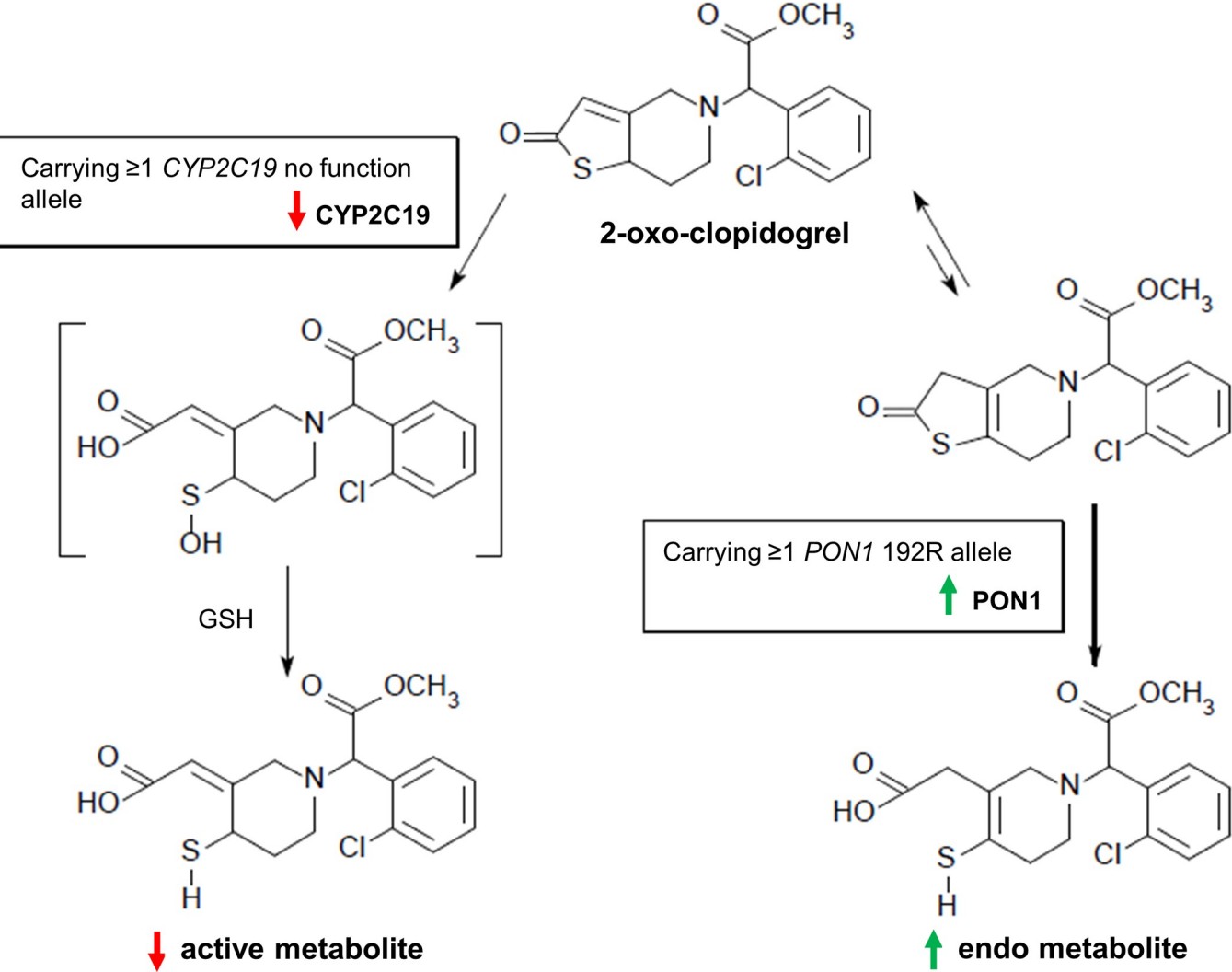

**Fig 1. Pathway involved in the formation of the active and endo metabolites of clopidogrel in patients with polymorphisms in cytochrome P450 family 2 subfamily C member 19 (*CYP2C19)* and paraoxonase-1 (*PON1*).** In carriers of ≥1 *PON1* 192R allele, high PON1 activity may accelerate an increase in the production of the endo metabolite of clopidogrel. Low CYP2C19 activity in carriers of ≥1 *CYP2C19* no function allele may reinforce this cascade via the delayed metabolism of 2-oxo-clopidogrel into an active metabolite, which allows PON1 to metabolize 2-oxo-clopidogrel into the endo metabolite of clopidogrel. GSH, glutathione.

In addition to hematocrit, this study revealed clinical and biochemical parameters including sex, lower platelet count, and higher total cholesterol level that were also associated with HPR by original and corrected PRU with clopidogrel. A recent report demonstrated that lower platelet count was associated with larger mean platelet volume, which reflected greater pro-thrombotic activity [32]. Another report demonstrated an association between total cholesterol level and clopidogrel resistance [33]. However, these associations remain controversial and the mechanism is unclear. Further study with larger samples might provide reliable conclusions.

This study had several limitations. First, this study did not include genetic polymorphisms other than *CYP2C19*2, *3*, and *17*, and *PON1* Q192R on post-clopidogrel platelet reactivity. Notably, *PON1* L55M, which lowers the activity of PON1 [34], and has an allele frequency of 3.4% in East Asians [21], was not included in the analysis. Moreover, in addition to the partial role of CYP2C19, many other enzymes are involved in the metabolism of clopidogrel, and the

role of PON1 could be relatively minor. Therefore, this study cannot exclude alternative results that might be related to other genetic polymorphisms. Second, the study was performed at a single center with a relatively small number of patients with limited clinical and laboratory information, which may have introduced some selection bias. Third, previous studies showed a PRU of ≥208 predicted thromboembolic events following pipeline embolization device placement for IA [17]; however, we could not systematically evaluate associations between periprocedural thromboembolic complications and HPR due to the heterogeneity of endovascular procedures and additional treatment for HPR (e.g., some patients with HPR received additional antithrombotic agents before and after treatment). Fourth, the lack of a multiple-comparison correction might have led to false-positive results. Finally, participants in this study were mostly Japanese, which might limit the generalizability of the findings.

## Conclusions

Alongside clinical, hematological, and biochemical parameters and the presence of ≥1 *CYP2C19* no function allele, carrying ≥1 *PON1* 192R allele is associated with HPR by original and corrected PRU with clopidogrel in patients undergoing elective neurointervention, although alternative results that might be related to other genetic polymorphisms cannot be excluded.

## Supporting information

**S1 File.**
(XLSX)

## Acknowledgments

We thank Emily Woodhouse, PhD, and J. Ludovic Croxford, PhD, from Edanz (https://jp.edanz.com/ac) for editing a draft of this manuscript.

## Author Contributions

**Conceptualization:** Koji Tanaka, Shoji Matsumoto.

**Data curation:** Gulibahaer Ainiding, Hidehisa Nishi, Tetsuya Hashimoto, Tsuyoshi Ohta, Nobutake Sadamasa, Ryota Ishibashi, Masanori Gomi, Makoto Saka, Haruka Miyata, Sadayoshi Watanabe, Takuya Okata, Kazutaka Sonoda, Junpei Koge, Kyoko M. Iinuma, Konosuke Furuta, Izumi Nagata, Keitaro Matsuo.

**Formal analysis:** Koji Tanaka.

**Funding acquisition:** Shoji Matsumoto.

**Methodology:** Kyoko M. Iinuma.

**Supervision:** Jun-ichi Kira.

**Writing – original draft:** Koji Tanaka.

**Writing – review & editing:** Shoji Matsumoto, Ichiro Nakahara, Tsuyoshi Ohta, Takuya Matsushita, Noriko Isobe, Ryo Yamasaki, Jun-ichi Kira.

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
