## [Decision Letter · Decision Letter 0]

11 May 2021

PONE-D-21-03186

*PON1* Q192R is associated with high platelet reactivity with clopidogrel in patients undergoing elective neurointervention: a prospective single-centre cohort study

PLOS ONE

Dear Dr. Kira,

Thank you for submitting your manuscript to PLOS ONE. After careful consideration, we feel that it has merit but does not fully meet PLOS ONE’s publication criteria as it currently stands. Therefore, we invite you to submit a revised version of the manuscript that addresses the points raised during the review process.

The manuscript should be revised  according to the criticisms raised by the Reviewer. In particular, it is reccomended that the Authors include also the L55M polymrophism in their analysis. The manuscript should be revised also including a discussion about the  minor role of PON1 in clopidogrel metabolism.

Please submit your revised manuscript by August 24, 2021. If you will need more time than this to complete your revisions, please reply to this message or contact the journal office at plosone@plos.org. Please include the following items when submitting your revised manuscript:

We look forward to receiving your revised manuscript.

Kind regards,

Cinzia Ciccacci

Academic Editor

PLOS ONE

Journal Requirements:

Reviewers' comments:

Reviewer's Responses to Questions

**Comments to the Author**

1. Is the manuscript technically sound, and do the data support the conclusions?

Reviewer #1: Yes

2. Has the statistical analysis been performed appropriately and rigorously? 

Reviewer #1: Yes

3. Have the authors made all data underlying the findings in their manuscript fully available?

Reviewer #1: Yes

4. Is the manuscript presented in an intelligible fashion and written in standard English?

Reviewer #1: Yes

5. Review Comments to the Author

Reviewer #1: This is a single-centre study evaluating the role of PON1 Q192R polymorphisms on the effect of clopidogrel in neurointervened patients.

Major comments:

The authors have not included in the analysis other genes and polymorphisms that could be of interest, so this study have a very large limitation in terms of results.

The authors should analyze at least the PON1 L55M polymorphism which reduces enzyme activity and is present in 5% of the Japanese population.

The result they present is an influence of the 192R allele on platelet aggregation, showing a worse response to clopidogrel in patients who carry it. The study by Bouman et al (ref 8) finds the opposite, but the authors do not mention it in the discussion.

It is important to note that 85% of the prodrug is metabolized by CES1 and only the remaining 15% will be metabolized to form the active metabolite, with CYP2C19 being responsible for the formation of 45% of 2-oxo-clopidogrel and 21% of the active metabolite. Bearing in mind that many other enzymes are also involved, the role of PON1 could appear to be minor. The authors do not comment on this.

The authors do not comment on the discussion of the only article that analyzes PON1 in neurointerventional patients, also analyzing many more genes and polymorphisms (Saiz-Rodríguez et al, ref 18). In that article no association with PON1 was found.

Authors state that they include the use of proton pump inhibitors, almost 75% of patients were receiving a PPI, which have been highly linked to clopidogrel response, via inhibition of CYP2C19 (Saiz-Rodríguez et al, ref 18), Saab et al. Ther Clin Risk Manag. 2015 Sep 23;11:1421-7, but they found no association? It’s doubtful.

Did the authors applied any multiple test correction?

Minor comments:

Loss-of-function and gain –of-function terminology is obsolete. Please change accordingly throughout the manuscript base on the CPIC standardization of terms (Caudle et al.)

Enrollment finished on 2015, why? Can authors enhance the sample size by including patients from 2015 to nowadays?

I suggest that genomic data and frequencies are compared with 1000 genomes project, which gives a different % for the analysed SNPs in the East Assian population.

Moreover, when first used, the abbreviation of DAPT should appear.

Line 174-178. The sentence should be rephrased, too long and difficult to follow.

6. PLOS authors have the option to publish the peer review history of their article (what does this mean?). If published, this will include your full peer review and any attached files.

Reviewer #1: No

---

## [Author Response · Author response to Decision Letter 0]

9 Jun 2021

Please see the attached "cover letter" and "response to reviewers" files. Thank you.

---

## [Decision Letter · Decision Letter 1]

21 Jun 2021

*PON1* Q192R is associated with high platelet reactivity with clopidogrel in patients undergoing elective neurointervention: a prospective single-centre cohort study

PONE-D-21-03186R1

Dear Dr. Kira,

We’re pleased to inform you that your manuscript has been judged scientifically suitable for publication and will be formally accepted for publication once it meets all outstanding technical requirements.

Kind regards,

Cinzia Ciccacci

Academic Editor

PLOS ONE

Additional Editor Comments (optional):

Reviewers' comments:

Reviewer's Responses to Questions

**Comments to the Author**

1. If the authors have adequately addressed your comments raised in a previous round of review and you feel that this manuscript is now acceptable for publication, you may indicate that here to bypass the “Comments to the Author” section, enter your conflict of interest statement in the “Confidential to Editor” section, and submit your "Accept" recommendation.

Reviewer #1: All comments have been addressed

2. Is the manuscript technically sound, and do the data support the conclusions?

Reviewer #1: Partly

3. Has the statistical analysis been performed appropriately and rigorously? 

Reviewer #1: No

4. Have the authors made all data underlying the findings in their manuscript fully available?

Reviewer #1: Yes

5. Is the manuscript presented in an intelligible fashion and written in standard English?

Reviewer #1: Yes

6. Review Comments to the Author

Reviewer #1: (No Response)

7. PLOS authors have the option to publish the peer review history of their article (what does this mean?). If published, this will include your full peer review and any attached files.

Reviewer #1: No

---

## [Editor Report · Acceptance letter]

28 Jul 2021

PONE-D-21-03186R1 

*PON1* Q192R is associated with high platelet reactivity with clopidogrel in patients undergoing elective neurointervention: a prospective single-center cohort study 

Dear Dr. Kira:

I'm pleased to inform you that your manuscript has been deemed suitable for publication in PLOS ONE. Congratulations! Your manuscript is now with our production department. 

Kind regards, 

on behalf of

Dr. Cinzia Ciccacci 

Academic Editor

PLOS ONE